# Elevated olivine weathering rates and sulfate formation at cryogenic temperatures on Mars

Paul B. Niles[1], Joseph Michalski[2], Douglas W. Ming[1] & D.C. Golden[3]

Large Hesperian-aged (~3.7 Ga) layered deposits of sulfate-rich sediments in the equatorial regions of Mars have been suggested to be evidence for ephemeral playa environments. But early Mars may not have been warm enough to support conditions similar to what occurs in arid environments on Earth. Instead cold, icy environments may have been widespread. Under cryogenic conditions sulfate formation might be blocked, since kinetics of silicate weathering are typically strongly retarded at temperatures well below 0 °C. But cryo-concentration of acidic solutions may counteract the slow kinetics. Here we show that cryo-concentrated acidic brines rapidly chemically weather olivine minerals and form sulfate minerals at temperatures as low as −60 °C. These experimental results demonstrate the viability of sulfate formation under current Martian conditions, even in the polar regions. An ice-hosted sedimentation and weathering model may provide a compelling description of the origin of large Hesperian-aged layered sulfate deposits on Mars.

---

[1] Astromaterials Research and Exploration Science Division, NASA Johnson Space Center, Houston, TX 77058, USA. [2] Department of Earth Sciences and Laboratory for Space Research, University of Hong Kong, Hong Kong, China. [3] ESCG, Houston, TX 77058, USA. Correspondence and requests for materials should be addressed to P.B.N. (email: paul.b.niles@nasa.gov)

The nature of the Martian climate through history remains an extremely important and unresolved question in planetary science. Despite huge amounts of data returned from orbiters and landers on Mars, there still remains uncertainty about whether the Martian surface environment ever supported an Earth-like climate and if so, for how long. Certainly, substantial geomorphological evidence has revealed that water has flowed across much of the Martian surface. However, this evidence is generally insufficient to further constrain the timescales involved, and many of the channels, deltas, lakes, and other features could represent only brief water activity[1, 2].

Substantial oxidized sulfur in Martian soils was detected by the Viking landers and has been suggested to form via oxidation of sulfide minerals or via reactions with acidic aerosols in the atmosphere[3–6]. However, more recent rover and orbiter results have revealed the presence of large deposits of layered sulfate-rich sediments (up to 30% SO₃) that largely occur within Hesperian-aged deposits on the Martian surface. On Earth, large deposits of sulfate-rich sediments typically form from bodies of liquid water and therefore their presence on Mars has been taken as evidence for widespread (though transient) lacustrine environments (~3.7 Ga)[7]. Yet, massive deposits of relatively young sulfates are also observed around the north pole[8], in an environment that is very unlikely to have involved surface water. If it can be shown that sulfates can form under the cold, dry conditions in the Martian polar regions on Mars in the Amazonian, it should force a serious reconsideration of how equatorial sulfate-bearing sediments formed in the Hesperian. Understanding the aqueous

environments which produce sulfate minerals on Mars is important to understand the evolution of the Martian climate.

In this paper we consider an alternate, more uniformitarian view of the ancient Martian climate, contending that prolonged warm temperatures were never present except in the earliest stages of its history, and that the atmosphere and climate have been similar to modern conditions throughout most of the planet's history[1, 9]. In this model, the formation of layered sulfate deposits represents a period of intense volcanic outgassing in a cold, dry climate where outgassed sulfur was concentrated into icy ash/dust deposits[10, 11]. Mass balance calculations have shown that sufficient SO₂ was degassed during this period to form massive sulfate-rich deposits[11]. In particular, it has not been clear how sulfate minerals could form at temperatures below 0 °C where the very low kinetics might effectively lower the weathering rate of basaltic minerals to a virtual standstill[12, 13]. Very slow dissolution rates have been measured during experimental dissolution of olivine and basaltic glass at temperatures as low as −19 °C in a CaCl₂–NaCl–H₂O brine[13]. However, at temperatures below 0 °C acidic solutions can become increasingly concentrated[14] through ice formation, which may actually enhance acid-weathering despite the slower kinetics at such low temperatures suggesting that sulfate formation may be possible at temperatures below 0 °C. We have conducted a set of experiments to test this hypothesis and show that the weathering rates of olivine at temperatures below 0 °C are sufficient to form sulfates in this cold, liquid water limited environment over timescales < 1000 years.

## Results

**Acid weathering experiments.** We performed laboratory experiments to simulate weathering of olivine by thin films of acid fluids at temperatures between −40 and −60 °C. The experiments subjected fine grained (5–53 μm) olivine particles to small amounts of 0.5 M sulfuric acid mixed with 400 μm silica beads at −40 and −60 °C in order to form a 10 μm liquid film coating each bead. These experiments were intended to simulate the exposure of small dust grains to thin films of sulfuric acid at cryogenic temperatures inside ice deposits on the surface of Mars. The longest duration experiments (12 days) yielded sulfate minerals even at temperatures, as low as −60 °C (Fig. 1). While these experiments intentionally and artificially induce contact between the mineral grains and acidic fluids, it is likely that this will occur naturally on Mars, where dust grains in the atmosphere will serve as nucleation points for ice and acidic aerosols which will ensure close contact between the reactants[15].

**Olivine dissolution rates and activation energy.** After quenching the experiment with a sodium acetate buffer, the concentrations of dissolved $Mg^{2+}$ and $Fe^{2+}$ were measured (Fig. 2, Supplementary Tables 1–4) and used to calculate dissolution rates by using a disappearing sphere model for a situation where substantial dissolution is expected and corrected with results from blank experiments[16]. Calculated rates of olivine dissolution from olivine in this work were $6.7(\pm1.2) \times 10^{-13}$ mol cm$^{-2}$ s$^{-1}$ for −40 °C experiments, and $2.2(\pm0.7) \times 10^{-13}$ mol cm$^{-2}$ s$^{-1}$ for the −60 °C experiments (Supplementary Figs. 1 and 2). The rate measured at −40 °C is comparable to forsterite dissolution rates in a pH 2.5 solution at 25 °C[17] suggesting that acid weathering under subzero conditions may be quite efficient in dissolving olivine (Fig. 3). This rate can also be compared with other laboratory studies of olivine dissolution using the Arrhenius equation

$$\ln k_{\mathrm{T}} = \ln k_{\mathrm{T,R}} - \frac{E_{\mathrm{a}}}{R}\left(\frac{1}{T} - \frac{1}{T_{\mathrm{R}}}\right) \qquad (1)$$

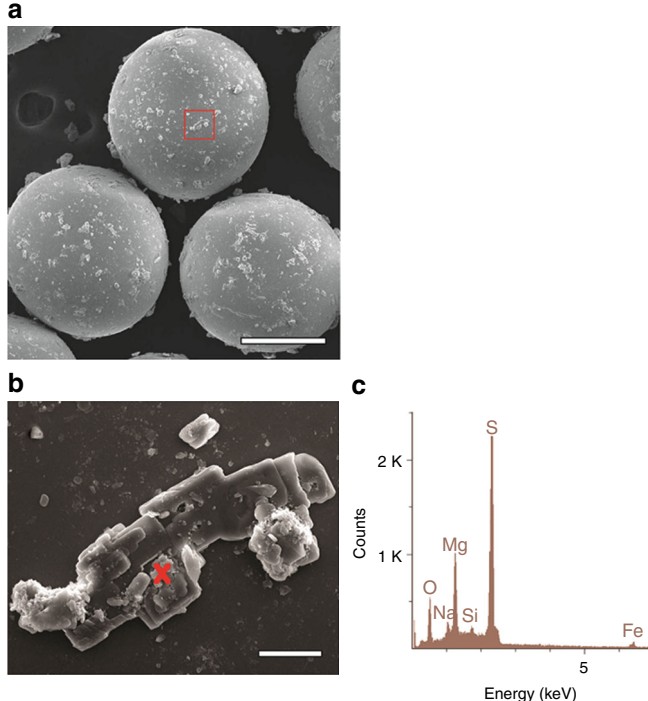

**Fig. 1** SEM secondary electron images and EDS spectra of experimental products recovered after 12 days at −60 °C. Recovery was accomplished by freeze drying rather than quenching with NaOH solution (see Methods: Quenching Procedure and Analysis). **a** Run products are located on 400 μm glass spheres used in experiments. Scale bar is 200 μm **b** Magnified view of mineral grain identified with red box in 1A. Red cross indicates location of EDS measurement. Scale bar is 10 μm **c** Energy Dispersive X-ray Spectroscopy (EDS) data that show substantial enrichments in Mg and S consistent with the presence of a Mg-sulfate mineral

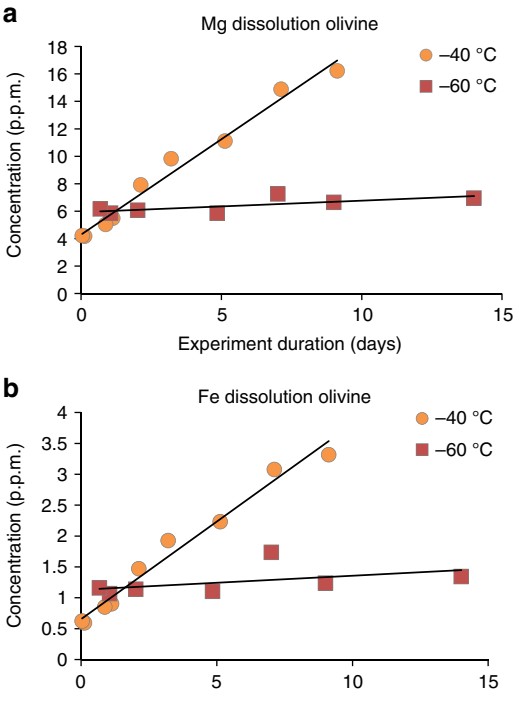

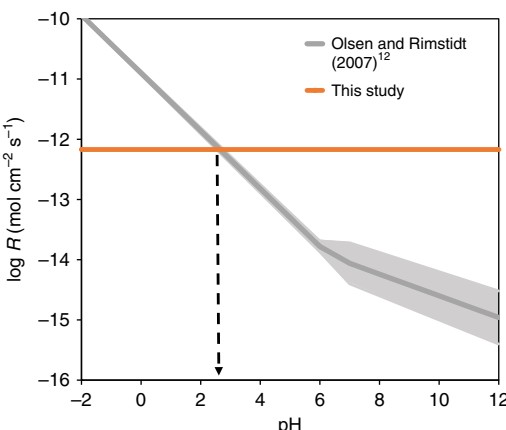

**Fig. 3** Relationship of weathering rates of olivine at 25 °C with pH. Weathering rates at −40 °C calculated in this study are shown by the orange bar. Dark gray lines indicate fit to the accumulated olivine dissolution data calculated by Olsen and Rimstidt[12] including uncertainties (s.e.m.) (light gray area) and are extrapolated below pH 1. Weathering rates measured in this study are equivalent to the weathering rate of forsterite at 25 °C and a pH of ~2.5[12]

**Fig. 2** Measurements of **a** $Mg^{2+}$ and **b** $Fe^{2+}$ concentrations in solutions after experiments were concluded. Linear best fit lines are shown for clarity although weathering rates were only calculated using longest duration experiments (> 3 days). The $R^2$ for the linear fits are 0.98 for the experiments at −40 °C. The $R^2$ values for linear fit of the −60 °C are not as good 0.53 for $Mg^{2+}$ and 0.22 for $Fe^{2+}$ as these experiments were closer to the blank values. Atomic absorption measurements were performed in triplicate with analytical uncertainties (s.e.m.) < 1% (< 0.05 p.p.m.) so that error bars are smaller than the data points. However the experimental procedure is not perfectly consistent, therefore we expect some natural variance in the results that would exceed the analytical uncertainties due to differences in how the materials interact during each experiment

where $k_T$ is the rate constant at temperature (233 K), $k_{T,R}$ is the rate constant at the reference temperature ($T_R = 298$ K), $E_a$ is the activation energy, and $R$ is the universal gas constant. There are substantial data available for all of these terms in the literature except for $k_{T,R}$ which is not well known at pH of −0.66 (expected pH of fluids at −40 °C, see materials and methods below). We can obtain a plausible $k_{T,R}$ by extrapolating the equation for forsterite dissolution rate formulated by Olsen and Rimstidt[12] to pH of −0.66 (−40 °C) and pH −0.73 (−60 °C). This results in a $k_{T,R}$ of $2.62 \times 10^{-11}$ mol cm$^{-2}$ s$^{-1}$ at −40 °C and $2.82 \times 10^{-11}$ mol cm$^{-2}$ s$^{-1}$ at −60 °C. Finally, we calculate the activation energy of forsterite dissolution to be 33 kJ mol$^{-1}$ (−40 °C) and 30 kJ mol$^{-1}$ (−60 °C). This is lower than what has been observed in other studies who have a compiled 'best estimate' of 63 kJ mol$^{-1}$ for forsterite dissolution[12, 13]. However this is very similar to values measured for fayalite dissolution under a range of temperatures (22.1 to −19.4 °C)[13]. The calculated activation energy of ~32 kJ mol$^{-1}$ under our experimental conditions suggests that olivine dissolution at cryogenic temperatures and very low pH may be very different than dissolution reactions in more dilute solutions at warmer temperatures. The difference may also simply be a product of the difficulty in extrapolating $k_{T,R}$ values to the extremely low pH values expected.

The experimental results also show that the ratio of Mg/Fe leaching rates proceed in stoichiometric fashion similar to the results from previous experiments at higher temperatures, where acidic solutions are shown to leach cations from the structures of the minerals in nearly stoichiometric fashion[18, 19], especially from finer grained material which is less susceptible to formation of weathering rinds[19, 20]. The observed stoichiometric release of $Fe^{2+}$ into solution was observable despite the fact that the experiments were conducted with terrestrial air. This indicates that the experimental conditions inhibited exchange between the oxygen in the air and the $Fe^{2+}$ in solution. This could have been the result of the low pH solutions or isolation from the air by ice formation. In this system, the dissolution capability of the acidic solution is maximized through eutectic freezing in an environment where the silicate minerals are extremely fine grained and have high surface areas. This provides an ideal environment for sulfate formation despite the very low temperatures which may be perceived to hinder dissolution rates.

## Discussion

The weathering rates measured in this study suggest that fine grained olivine on Mars at the surface will be weathered over short timescales if it is exposed to $H_2SO_4$ aerosols at temperatures warmer than −40 °C. It is clear that even the extremely cold modern surface temperatures can typically exceed −40 °C[21]. In addition, we show evidence that this dissolution process, if followed by sublimation, would result in sulfate minerals (Fig. 1, Supplementary Fig. 3, Supplementary Table 5) as should be expected given the chemistry involved. Using the dissolution rate measured in this study at −40 °C, a 100 μm olivine grain would have a lifetime of ~5 yr[12], however it is likely that this laboratory derived dissolution rate is much higher than one would expect to see in the field[18]. For example, salts may accumulate in the solutions which may hinder dissolution rates[13]. Nevertheless, even an decrease of several orders of magnitude (> 1000) in the dissolution rate makes olivine unstable on 1000 to 10,000 year periods, which is short compared to the hundreds of millions of years available during the Hesperian.

Sulphates have been found in ice deposits in Greenland and Antarctica on Earth that have been attributed to forming within the ice deposit[22–25]. Mg, Na, and Ca sulfates have all been found in micro-inclusions in the ice[23]. These have been attributed to interaction between acidic aerosols and sea salt or airborne

dust[23]. The temperatures at which these reactions occurred are not well constrained. Likewise, weathering has been observed in soils in the Antarctic Dry Valleys indicating possible cryogenic silicate weathering processes[26]. While these terrestrial examples are promising, the active hydrological cycling on Earth makes it difficult to observe these processes on the larger scales on which it may have operated on Mars.

If cryogenic sulfate formation was active on Mars in the past, it should also be active in the polar regions of Mars today—although this process would be strongly controlled by the amount and rate of volcanic outgassing of sulfur. The modern polar regions of Mars have large deposits of ice and dust that have been deposited over long periods of time and have resulted in vast, layered sedimentary deposits. The results of this study suggest that this environment should be conducive to formation of sulfate minerals provided a sufficient amount of sulfur-rich volatiles were supplied during a given period.

Indeed, large regions of sulfate-rich material have been detected on and around the modern north polar region of Mars and there is evidence that the material is derived from the polar sediments themselves[27]. It has been shown that even in the polar regions, conditions could be warm enough to melt pure water ice around dark dust grains on steep equatorial facing slopes in the polar region[28]. Our experiments demonstrate that weathering occurs at even much lower temperatures than the 273 K, and therefore if $H_2SO_4$ aerosols are present we expect weathering to occur over an even larger area and for longer portions of the day. However, the dominant sulfate mineral identified in the polar regions is $CaSO_4$, which would have been derived via weathering of minerals other than olivine, and Ca-bearing phases were not included in this work.

The experimental evidence combined with field evidence from both the Earth and Mars provides a compelling argument that cryogenic sulfate formation is not only possible, but would likely be unavoidable on Mars. The evidence clearly shows that cryogenic weathering can operate, even under the extremely cold current Martian conditions. Furthermore, the combination of extensive volcanic outgassing and obliquity variations in the Hesperian[29, 30] provides a plausible mechanism for the deposition of large amounts of dust, ice, and sulfuric acid aerosols in the equatorial regions. During certain periods of high orbital obliquity the polar regions received greater solar insolation than the equatorial regions and would almost certainly result in deposition of ice at lower latitudes[30]. This provides a means for depositing ice/dust/sulfur deposits at equatorial latitudes in a process similar to that which deposits the polar layered deposits today[10]. Indeed there exists substantial geomorphic evidence of glacial activity in the equatorial regions in the areas, where the layered sulfate deposits are found[31, 32]. Given the frequency and magnitude of the obliquity variations, a substantial sedimentary record could have been produced by this process[33]. On the basis of the abundance of layered sediments in the equatorial regions, estimates for the amount of sulfur required to form these deposits is well within estimates of $SO_2$ outgassing in the Noachian and Hesperian[11]. The source of the cations required to form these sulfate minerals may not have come exclusively from olivine weathering, but the preponderance of Mg-, Fe-rich sulfates in these deposits suggests that olivine was an important source. The scenario described here represents a completely different weathering regime from anything preserved in the terrestrial geologic record and it is enabled by volcanism in a subfreezing environment on a planet without an ocean.

## Methods

**Experimental materials and setup**. In cryogenic experiments, 0.5 mL of 0.5 M $H_2SO_4$ was pipetted into 40-mL freezer-proof polycarbonate tubes containing 10 g of acid-washed 400-μm silica beads and frozen at a predetermined temperature (−40° and −60 °C) in a precisely temperature-controlled freezer. In total 50 mg of San Carlos olivine (Fo90) ground under acetone (5–53 μm) were then added on top of the acid-treated silica beads, and the tube was removed from the freezer and shaken vigorously using a vortex mixer for 1 min to mix the reactants and immediately transferred back to the freezer. The amount of dilute (0.5 M) acid was calculated to be just enough to coat the silica beads to a ~10 μm thickness. On the basis of modeling results by Marion[14] we expect that at the experimental temperatures the films of acid solution would be mixtures of ice and cryo-concentrated acidic solutions of 4.8 molal at −40 °C (pH = −0.66) and 5.6 molal at −60 °C (pH = −0.73). The silica beads provided a matrix on which the reaction could take place providing surface area, which could be coated with a thin layer of solution, and could evenly distribute the olivine grains while maintaining low temperatures. If the beads were not used, then it would be very difficult to get olivine grains mixed with an icy acid solution at cryogenic temperatures. The glass beads provided the momentum to break up the semi-frozen solutions and to get reactants fully mixed. The beads also help prevent contact between the acidic films and the olivine grains. By limiting the amount of time outside the freezer to only 1 min, temperatures were not allowed to increase substantially inside the tubes, although this could not be measured directly.

**Quenching procedure and analysis**. At the conclusion of each experiment, reaction products were collected by adding 10 mL of 0.1 M sodium acetate buffer (pH = 7) to neutralize remaining acidity and to melt all the ice. The quenched mixture was shaken for 30 s and filtered immediately using 0.2 μm Nalgene filter unit under vacuum to separate the supernatant. The elements Mg and Fe in buffered extracts were determined by using a Perkin Elmer Analyst 800 atomic absorption spectrometer with AS800 Autosampler using matrix matched reference standards. Si was not measured because of extremely high backgrounds provided by the Si-spheres. Experiments were run for different durations ranging from 1 h to 9 days. Select experiments were not quenched with NaOH solution, but were instead taken out of the freezer and immediately freeze dried. The products were then mounted for scanning electron microscope (SEM) imaging.

**Dissolution rate calculation**. The measured solution compositions were converted into dissolution rates by using a disappearing sphere model for a situation where substantial dissolution is expected[16]:

$$n = 4\pi k S I \left( r_o^2 t - kVt^2 r + \frac{k^2 V^2 t^3}{3} \right) \qquad (2)$$

Where, $n$ is the cumulative number of moles (Mg+Fe) released from a sphere of radius $r$, $k$ is the rate of dissolution of the mineral in moles $m^{-2} s^{-1}$, $V$ is the molar volume of the mineral, $t$ is the time of dissolution, $S$ is the number of particles, and $I$ is the correction factor to account for the atoms per formula unit (e.g., 0.5 for forsterite $Mg_2SiO_4$). Initial sphere radius ($r_o$) for olivine was calculated to be 8 μm assuming a log-normal distribution of particle sizes between 5 and 53 μm[34]. S was calculated using the initial sphere radius (olivine particles) and the mass of the sample which did not include the silica spheres. Because Eq. (2) cannot be easily solved for $k$, a method of successive approximations was used to calculate the number of moles until it matched the experimental results to four significant digits. The final dissolution rate was calculated by averaging the rates calculated from the longest duration experiments (>3 days). The rates were also corrected for results from blank experiments conducted with no olivine present. The blank experiment consisted of adding all of the materials except for the olivine and immediately adding the buffer solution and collecting the product (<1 h total duration). The short duration blank did show a significant background level (above 1 μmol) of Mg and Fe (Supplementary Table 1). In addition, similar experiments run at 7 days length showed minimal additional contribution of Mg and Fe (~1 μmol) from silica spheres (Supplementary Table 4). Acid-free blanks (all materials except for acid) were also run using similar durations (Supplementary Table 3). Assuming 50 mg of olivine and 0.5 mL of 0.5 M $H_2SO_4$ solution, we can calculate 0.25 mmol of $H_2SO_4$ and 0.54 mol of olivine. As 2 mol of $H_2SO_4$ are neutralized by 1 mol of olivine, the amount of olivine in the experiments is double the amount needed to neutralize the acid. This is consistent with what would be expected in natural systems, where the amount of silicate should overwhelm the available acidity.

**Data availability**. All data generated or analyzed during this study are included in this published article (and its supplementary information files).

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

## Acknowledgements

We would like to acknowledge funding from Johnson Space Center.

## Author contributions

P.B.N. conceived of the study, helped reduce the data, and wrote most of the manuscript. J.M. and D.W.M. wrote some portions of the manuscript. D.C.G. performed the experiments and analyses and reduced the data.

## Additional information

**Competing interests:** The authors declare no competing financial interests.

