## [Peer Review File · Nature Communications]

Reviewers' Comments:

Reviewer #1:

Remarks to the Author:

These are very interesting and novel experiments with very interesting implications.

My main concern is the use of the BET surface area rather than the geometric surface area to calculate the particle sizes (Line 235 of the manuscript). The ratio of the BET surface area to the geometric surface area is generally quite large (it can be ~ 10 for olivine). Therefore, I don't think that using the BET surface area is an acceptable estimate, but that the geometric surface area should be calculated by dividing the BET surface area by the roughness (~ 10 and available in the literature), and then using that surface area. I completely understand that in quantifying processes, assumptions must be made, but in this case, since it is possible to make a much better assumption, the authors should do that. It is also not very much work – simply using a different number in the equation on line 235 of the manuscript.

Minor points:

Lines 120-122 – Why do you think that the lower rates observed at -60 deg C are simply not due to slower rates at lower temperatures?

Lines 122-125 – I think that it is confusing to refer to your lower temperature experiments as being blanks when you also have actual experimental blanks.

Reviewer Comments:

Authors Responses in Bold

Reviewer #1 (Remarks to the Author):

These are very interesting and novel experiments with very interesting implications.

My main concern is the use of the BET surface area rather than the geometric surface area to calculate the particle sizes (Line 235 of the manuscript). The ratio of the BET surface area to the geometric surface area is generally quite large (it can be ~ 10 for olivine). Therefore, I don't think that using the BET surface area is an acceptable estimate, but that the geometric surface area should be calculated by dividing the BET surface area by the roughness (~ 10 and available in the literature), and then using that surface area. I completely understand that in quantifying processes, assumptions must be made, but in this case, since it is possible to make a much better assumption, the authors should do that. It is also not very much work – simply using a different number in the equation on line 235 of the manuscript.

Minor points:

Lines 120-122 – Why do you think that the lower rates observed at -60 deg C are simply not due to slower rates at lower temperatures?

Lines 122-125 – I think that it is confusing to refer to your lower temperature experiments as being blanks when you also have actual experimental blanks.

We have recalculated the particle size using geometric surface area instead of the BET surface area as described by the reviewer. This is now described in the manuscript. This resulted in a change from 1.7 um diameter to 8 um diameter. This change required a recalculation of the weathering rates, which became faster/larger. Thus this did not have an effect on the conclusions. In the course of the recalculations, we also revisited the activation energy calculation and revised those numbers as well.

We have also removed the language that was on lines 120-125 about -60 degree experiments. We agree with the reviewer.

Again we thank all reviewers for their thorough comments and hard work to improve this manuscript.